# Decoding the Role of CD271 in Melanoma

**DOI:** 10.3390/cancers12092460

**Published:** 2020-08-31

**Authors:** Anna Vidal, Torben Redmer

**Affiliations:** Department of Biomedical Sciences, University of Veterinary Medicine, Institute of Medical Biochemistry (HA/I), Veterinärplatz 1, 1210 Vienna, Austria; anna.vidal@vetmeduni.ac.at

**Keywords:** CD271, melanoma, neural crest stem cells, migration, metastasis

## Abstract

The evolution of melanoma, the most aggressive type of skin cancer, is triggered by driver mutations that are acquired in the coding regions of particularly BRAF (rat fibrosarcoma serine/threonine kinase, isoform B) or NRAS (neuroblastoma-type ras sarcoma virus) in melanocytes. Although driver mutations strongly determine tumor progression, additional factors are likely required and prerequisite for melanoma formation. Melanocytes are formed during vertebrate development in a well-controlled differentiation process of multipotent neural crest stem cells (NCSCs). However, mechanisms determining the properties of melanocytes and melanoma cells are still not well understood. The nerve growth factor receptor CD271 is likewise expressed in melanocytes, melanoma cells and NCSCs and programs the maintenance of a stem-like and migratory phenotype via a comprehensive network of associated genes. Moreover, CD271 regulates phenotype switching, a process that enables the rapid and reversible conversion of proliferative into invasive or non-stem-like states into stem-like states by yet largely unknown mechanisms. Here, we summarize current findings about CD271-associated mechanisms in melanoma cells and illustrate the role of CD271 for melanoma cell migration and metastasis, phenotype-switching, resistance to therapeutic interventions, and the maintenance of an NCSC-like state.

## 1. Introduction

The acquisition of a metastatic phenotype presents a hallmark in the progression of solid tumors and is rather marked by transcriptomic and epigenetic than genetic changes, mediating the activation or inactivation of molecular programs. The latter determine the progression stages of melanoma and the therapeutic control of tumors. Patients diagnosed with melanoma in non-metastatic stages IA and IIA show a five-year overall survival (OS) of 95.3 ± 0.4% and 78.7 ± 1.2%. The metastasis into regional lymph nodes and the formation of satellite skin metastases are observed in stage IIIA, resulting in a decrease of OS to 69.5 ± 3.7%. The systemic dissemination of melanoma cells throughout the body of stage IV patients leads to metastasis formation in distant organs particularly, lung, liver and brain which is associated with worst outcome and OS to 9.5 ± 1.1% [1]. The adaptation of tumor cells to prevailing microenvironmental conditions that are defined by the composition of growth factors and extracellular matrix (ECM) proteins is prerequisite for the proper establishment of metastases at distant organ sites [2,3]. Several lines of evidence suggest that metastasis is a sequential process [2,4] that likely select for a cellular subset featuring a high metastatic capacity and stem-like phenotype. The latter enables a rapid adaptation to different organ-specific conditions [5]. However, mechanisms driving the metastatic cascade in melanoma are still not well understood, although some promising candidates have been identified.

The nerve growth factor receptor CD271 (p75^NTR^, NGFR) was identified in 1968, functionally described in 1986, and associated with melanoma aggressiveness and metastasis in the late 1980s and early 1990s. Particularly, Herrmann et al. associated the expression of CD271 with increased invasiveness and brain metastasis of MeWo-70W cells [6,7]. However, CD271 was rediscovered only ~30 years later. In 2010, Boiko et al. established that CD271 labels a subset of melanoma-initiating cells (MICs, aka melanoma stem cells) capable of tumor formation and differentiation. Most strikingly, CD271 showed a mutually exclusive expression with typical melanoma cell surface markers like MART1 and HMB45 [8]. Hence, clinical trials targeting melanoma-specific antigens failed due to the insufficient targeting of MICs, which consequentially promoted tumor relapse. MICs that exhibit expression of CD271 are capable of renewing the tumor mass and give rise to differentiated progeny that feature typical melanocyte/melanoma antigens. Boiko et al. and others provided insight into the role of CD271 in melanoma but also demonstrated that the expression and localization of CD271 like other markers underlie cellular plasticity [8,9]. The expression profiling of melanoma cells with a stable knockdown of CD271 demonstrated for the first time that cellular properties of a subset of melanoma cells is regulated in a CD271-dependent fashion. Subsequent studies have shown that CD271 is part of a network controlling basic properties of melanoma cells. 

In the present review, we will summarize recent findings about the role of CD271 in melanoma and demonstrate that CD271 controls basic properties of melanoma cells, particularly migration and metastasis.

## 2. CD271 in Development

The expression of CD271 is observed at different stages of murine and human development, found earliest in cells of the inner cell mass of the murine blastocyst [10], maintained in murine embryonic stem cells (mESCs) but lost upon differentiation [11]. During vertebrate nervous system development, CD271 labels a large proportion of migrating multipotent neural crest stem cells (NCSCs) capable of self-renewal [12], which arise from the embryonic ectoderm. NCSCs delaminate from the neuroepithelium by epithelial-to-mesenchymal transition (EMT) and migrate along body axes. Migrating neural crest (NC) cells lose the expression of E-cadherin (CDH1) and gain expression of human natural killer-1 (HNK1)/3-beta-glucuronosyltransferase 1 (B3GAT1), CD271, and endothelin receptor type B (EDNRB) [13,14] (Figure 1a). The expression of CD271 enables the tracking of NCSCs during early human NC development and the isolation of multiple NCSCs from mammalian fetal peripheral nerve and differentiated human embryonic stem cells (hESCs) [12,13,15]. The hESC-derived NCSCs feature a high expression of CD271, SOX10, PAX3, etc. but a low expression of lineage-specific differentiation markers [15], reviewed in references [12,16]. Likely, the maintenance of multipotent NCSCs during NC development is driven by the transcription factor forkhead box d3 (FOXD3), which acts in concert with CD271 [17,18].

Following delamination, CD271^+^/SOX10^+^/HNK1^+^/CDH1^−^/FOXD3^−^ NCSCs migrate through the periphery and give rise to differentiated progeny; among them are muscle cells, glia cells, neurons, and melanocytes [19,20] (Figure 1a). The formation of melanocytes like other NCSC-derivatives is controlled by a network of regulatory transcription factors, particularly SOX10. The genetic ablation of SOX10 and CD271 in a mouse model system consistently led to a complete loss of the melanocyte-differentiation capacity and loss of the NC compartment [21,22]. The EMT that in turn is mediating the delamination of NCSCs from the NC (Figure 1a) is mediated by binding of the transcriptional repressors SNAI1 (Snail), SNAI2 (Slug), and ZEB1 to the E-cadherin/CDH1 promoter, reviewed by Barrallo-Gimeno et al. [23]. First insight into the functional role of CD271 in cell migration was provided by murine models, which enabled the conditional knockout of CD271 in NC cells. The latter revealed a decrease in the sciatic nerve diameter and a deficiency of hematopoiesis, reviewed by Wislet et al. [14]. Although the role of CD271 during melanocyte development and specification (Appendix A, comprehensively reviewed by Douarin et al., White et al. and Blake et al. [19,24,25] is unknown, knockdown studies in melanoma revealed a close relationship of CD271 with FOXD3, SOX2, and SOX10. This may suggest that CD271 promotes the maintenance of a NCSC-state in vivo and at least partially in vitro by the transcriptional control of NCSC-specifier. 

## 3. CD271 in the Skin and Development of Melanoma

CD271 is predominantly expressed in proliferating (Ki67^+^) and dormant (Ki67^−^) cells of the *stratum basale*, particularly melanocytes and transient-amplifying (TA) keratinocyte progenitors [26], (Figure 1b). In the latter, CD271 controls an early and dosage dependent step of keratinocyte differentiation [27]. Albeit the role of CD271 in melanocytes is not completely understood, several lines of evidence suggest that CD271 prevents UV-radiation-induced apoptosis in KIT (c-KIT) and MITF-positive melanocytes via up-regulation of the anti-apoptotic protein BCL-2 (Appendix A). The latter in turn is induced by bioactive NGF that is secreted by keratinocytes, which reside in close proximity to melanocytes. Moreover, levels of NGF and CD271 are increased in keratinocytes and melanocytes in vitro in response to UV-radiation, respectively [28,29,30]. Moreover, levels of CD271 expression are significantly higher in melanoma than in other types of skin cancer like basal cell carcinoma (BCC) and squamous cell carcinoma (SCC) (Appendix A), showing less severe progress. Besides skin melanocytes, a strong expression of CD271 is found in the eye´s choroid and in uveal melanoma (Appendix A). 

The formation of melanocytic nevi is the consequence of a hyper-activation of the MAPK pathway, triggered by acquired mutations in BRAF, most prevalently BRAF^V600E/K^ (~15–20% and ~40–50%) [31]. The constitutive activation of BRAF temporally leads to a focal proliferation of melanocytes, which consequently induces a cell cycle arrest and oncogene-induced senescence [32,33,34]. Hence, the loss of tumor suppressor genes, e.g., p16^INK4a^, PTEN, or TP53, is prerequisite for the transformation of BRAF-mutated melanocytes and formation of melanoma [35,36]. This genetic dependency was recapitulated in a murine model system first established by Dankort et al. In this system, oncogenic Braf^V600E^ was expressed under control of the tyrosinase/Tyr promoter and, hence, was restricted to melanocytes. Whereas the distinct expression of Braf^V600E^ triggered oncogene-induced senescence, the genetic loss of Pten mediated melanoma formation with a high efficacy [37]. Besides Braf, mutated Nras, Kras, or Hras were shown to effectively induce melanoma formation in combination with a genetic ablation of p16^Ink4a^, p53 and Cdkn2a among other, reviewed by Perez-Guijarro et al. [38]. In addition, activated AKT3 cooperates with BRAF^V600E^ to promote melanocyte transformation, which in turn is associated with a growth factor independent proliferation and enhanced anchorage-independent growth [33]. Although the AKT signaling pathway is frequently activated in melanoma progression, the loss of PTEN predominantly precedes the activation of AKT3 [39].

The identification of CD271 as a marker of MICs by Boiko et al. stimulated the intense investigation of the receptor in the field of melanoma and beyond. Most intriguingly, the expression of CD271 defines a subpopulation of melanoma cells lacking the expression of typical antigens like MART-1, HMB45, melanoma-associated antigens (MAGE) and regulators of proliferation or melanin synthesis e.g., MITF or TYR [8,40]. Likely, the low expression of MITF in CD271^+^ amelanotic tumors and cell lines is the consequence of a co-expression of CD271 and transcriptional repressors of MITF, e.g., FOXD3 and/or SOX2 [8,17,40,41,42]. The investigation of melanoma cell sub fractions retaining the lipophilic dye PKH26 revealed that even the pool of CD271^+^ cells contains proliferating (CD271^+^/Ki67^+^/PKH26^-^) and slow cycling/label-retaining (CD271^+^/Ki67^−^/PKH26^+^) sub sets [40]. The latter subset showed a higher expression of DNA-repair genes [43] and was linked with resistance to vemurafenib and a brain metastatic phenotype [44,45].

However, the persistence of CD271 expression in tumor cells is unknown and likely indeterminable. In addition, in vitro established tumor-derived melanoma cells exhibit an unstable and fluctuating expression of CD271 which might be stabilized by microenvironmental cues within the tumor and/or are potentially regulated by the circadian clock [46,47]. In addition, the level of CD271 increased in response to stress conditions, e.g., drug-induced DNA-damage [8,9,43,47,48]. Indeed, several melanoma metastases nearly containing ~90–100% of CD271 expressing cells were observed, suggesting the existence of a mechanism that stabilizes or forces CD271 expression in vivo. Consequentially, CD271 regulates key features of melanoma cells; particularly, a stem-like/NCSC-like phenotype and cell migration and plasticity abrogated in melanoma cells with a stable knockdown of CD271 (see Section 5). 

However, the role of CD271 in melanoma development and progression is not well defined. Recently, the comparative analysis of gene expression data of CD271^+^ and CD271^−^ melanoma cells and CD271^+^ melanocytes identified AKT3 (but not AKT1 or AKT2) among the top up-regulated genes serving as a key mediator of survival [49]. In addition, a pathway analysis revealed SHC1 as interaction partner of CD271 and increased levels of CAMKII in CD271^+^ cells serving as mediators of activated AKT3 [49]. Consistently, the stable shRNA-mediated knockdown of CD271 decreased the expression of AKT3 [50], suggesting that the concerted action of CD271 and AKT3 may promote the melanocyte transformation and maintenance of early established melanoma cells. However, whether CD271 serves as a crucial supportive factor controlling melanocyte transformation and melanoma progression remains unknown, although levels of CD271 expression increase with progression stages (Figure 1c,d). Only in vivo models will uncover the relevance of CD271 expression in this process. The comparative expression profiling of melanocytes and melanoma cells revealed that CD271 is associated with signaling processes involved in skin development, metabolic hormone processes, cell adhesion and ion channel activity [49]. In addition, the comparison of melanocytes with metastatic melanoma cells revealed a predominant expression of genes associated with interferon response and signaling via E-cadherin (CDH1) in melanocytes and genes involved in metastasis, EMT, and EGFR-signaling in melanoma (Appendix A).

## 4. Identification of CD271-Associated Genes

Over the past few years, it has been recognized that a subset of melanoma cells features an NCSC-like phenotype [8,40,51]. The stable or transient knockdown of CD271 in melanoma cells revealed that the NCSC-like phenotype is maintained by expression of CD271 (Figure 1d) which in turn controls the levels of downstream targets, e.g., SOX10, FOXD3, and SOX2 (Appendix A, not shown) [12,40,52]. The loss of CD271 consequentially induced a severe impairment of cell survival and proliferation, most likely caused by a loss of genomic integrity that in turn was associated with a reduced expression of DNA-repair genes [40,43] and revealed a mutually exclusive expression with inhibitors of metastasis, KISS1, and DMBT1 (Figure 2) [43]. In addition, the decrease of CD271 significantly reduced the capability of tumor formation in vivo and cell migration in vitro [40,44,50]. On the other hand, the overexpression of CD271 increased levels of SOX2 and RHOJ and the propensity of migration and metastasis of A375 cells [43,53]; however, the underlying mechanisms are poorly understood. These experiments strongly suggest a more comprehensive role of CD271, serving as determinant of melanoma cell properties independent from the presence of a BRAF^V600E^-mutation. The presence of the latter was not sufficient to prevent melanoma cells from apoptosis induced by the loss of CD271 expression [44]. Melanoma cell properties are likely controlled by a concerted cooperation of CD271 and SOX10 as the down regulation of either CD271 or SOX10 showed comparable effects [22]. Hence, CD271 and SOX10 share a common set of associated genes or targets, e.g., MITF, CDKN1A, TP53, and CCNA1 [40]. Interestingly, the loss of SOX10 reduced the level of CD271, suggesting a mutual dependency. Concordantly, the genetic ablation in Tyr::Nras^Q61K^ mice demonstrated that Sox10 is required for melanoma formation and proliferation, survival, and tumor maintenance [22]. However, if the genetic ablation of CD271 in melanocytes impairs melanoma formation has not yet been investigated.

## 5. CD271 in Migration, Metastasis, and Cellular Plasticity

The systemic dissemination of tumor cells throughout the body presents a hallmark of tumor progression and is still the leading cause of cancer-related death. Particularly the sequential formation of metastases at distant organs is associated with the loss of disease control and poor prognosis, as reviewed by Ackermann et al. [54]. Brain metastases (BM) are most challenging [55] and develop in 20–40% of melanoma cases during the course of disease, with unclear subtype specificity, as reviewed by Redmer [56]. The efficient systemic dissemination of melanoma cells relies on the maintenance of a migratory phenotype, which is likely inherited from NC cells [57,58]. Likely, the metastatic niche plays an important role for the maintenance of the migratory phenotype. In 1995, Marchetti et al. demonstrated that the ligands for neurotrophin receptors CD271 and tropomyosin-related kinases (TRKs, see Section 6), nerve growth factor (NGF), and neurotrophin 3 (NTF3) were expressed by niche cells within the brain but not brain metastatic melanoma cells [59]. On the contrary, the latter expressed CD271 and TRKC and featured migration at the invasion front likely in a paracrine response. The relevance of this dependency was demonstrated by Truzzi et al., who observed increased migration of metastatic melanoma cells after stimulation with NGF, NTF3, and NTF4 [60]. Although the in vitro situation might not directly reflect mechanisms of tumor cell migration within the brain, these studies suggest a role of neurotrophins in cell migration. More than a decade later, studies by Boiko et al. evidenced that the expression of CD271 mediates the stemness of melanoma cells and serves as a regulator of metastasis [8]. Later functional studies established that CD271 acts as a key regulator of the NCSC-like state in melanoma that, in turn, programs the migratory phenotype [40,50]. The expression profiling and gene-set enrichment analysis (GSEA) of melanoma cells with forced expression of CD271 revealed the association with genes conferring metastasis and relapse. Concordantly, live-cell imaging-based scratch-wound assays revealed that melanoma cells with a high endogenous level or forced expression of CD271 featured a significantly enhanced migration into the scratch wound than cells with a low expression and, consistently, the knockdown of CD271 decreased the migratory capacity [43,50]. Among the genes predicting either melanoma metastasis or relapse were hyaluronan mediated motility receptor (HMMR), NIMA related kinase 2 (NEK2), and DNA-repair genes. HMMR and NEK2 were associated with increased migration, metastasis, and poor survival in cancer [43,61,62,63,64,65,66,67,68], and Kauffmann et al. associated metastasis of melanoma with a high expression of DNA-repair genes and pathways [69]. We observed increased levels of DNA-repair genes RAD21, MSH6, and RAD51 in CD271^+^ but not CD271^−^ MACSed cells [43] hence, the DNA-repair capacity is potentially regulated in a CD271-dependent manner [43]. Mutant BRAF^V600^ serves as yet another driver of metastasis as mutant melanoma cells featured a higher endogenous migration capacity in vitro than wild type (wt) cells [43,70]. Although the level of CD271 in tumors is likely unstable [47], several processes raising the expression of CD271 have been identified in vitro. Particularly, cellular stress induced by therapeutic interventions [43,44,71], hypoxia [72,73], or inflammation [74,75,76,77] increased the expression of CD271. Chemotherapeutics have been used for more than 40 years in the first-line treatment of melanoma and are still in clinical use, particularly applied to patients lacking druggable mutations or patients who became refractory to immune checkpoint inhibitors [78]. However, DNA-damage inducing chemotherapeutic drugs increased the cell surface and total levels of CD271 after 24 hours of treatment in drug-sensitive (parental, Par) MeWo cells (Appendix A). The significant increase in the DNA-damage and p53-pathways (Appendix A) likely suggest a role of both regulating the levels of CD271 in vitro and in vivo. Besides mediators of the DNA-damage response, NGF and BDNF were up regulated in a short-term (24h) response to cisplatin (Appendix A), potentially triggering autocrine signaling processes e.g., migration. In addition, increased expression of CD271 and NGF was observed in chemo-resistant MeWo cells featuring increased migration [43]. Concordantly, the selective inhibitor of oncogenic BRAF^V600E/K^ vemurafenib mediated resistance via increased expression of CD271 [44]. 

In summary, therapeutic interventions select for highly migratory and metastatic melanoma cells, likely via the modification of levels of CD271. This statement is underpinned by the intriguing finding that the injection of a neutralizing antibody raised against CD271 blocks melanoma metastasis in a mouse model system [79]. 

Recently, the expression of CD271 was recognized as a regulator of phenotype switching, a process that enables the rapid and reversible conversion of non-stem-like into stem-like or proliferative into invasive states. The latter was associated with the AXL^high^/MITF^low^ program [53,54,55,56,57,58]. In addition, Hoek et al. specified the proliferative/low-migratory phenotype by (1) the levels of MITF-targets (TYR, DCT, PMEL, MLANA), (2) NC-related genes (SOX10, TFAP1A, EDNRB), and (3) the sensitivity toward the growth-inhibitory effect of TGF-β and the proportion of Ki67-positive cells [80,81]. Contrary, the migratory/low-proliferative phenotype was specified by EMT-inducers ZEB1, TWIST1, and c-JUN as well as CD271 and AXL [53,82]. The conditional overexpression of CD271 in melanoma cell lines impaired the proliferation in vitro and in vivo but enhanced metastasis formation at distant organs. Furthermore, Restivo et al. demonstrated that the manipulation of levels of CD271 was sufficient to control the switch between proliferation (“growing”) and invasion (“going”) of melanoma cells. Furthermore, Restivo et al. suggested that the interaction of CD271 with TRKA and the release of the CD271 intracellular domain (ICD) serve as responsible mechanisms, leading to the loss of adhesion via up regulation of cholesterol production and impaired proliferation [53]. The mechanism regulating the switch of CD271^high^ into MITF^high^ cells and vice versa is likely found in all organ-metastases and proceeds via a transient CD271^+^/MITF^+^ cell state. Consistently, the analysis of melanoma BM (MBM) revealed that CD271^high^ cells exhibit a lack of expression of MITF and MITF-targets [50] but feature a high enrichment of CD271-target/associated genes (study GSE50493 [83]) and genes associated with an NCSC-like phenotype and cell migration (Appendix A). 

Several mechanisms, which trigger and control cellular plasticity and phenotype switching, are plausible, besides CD271-ICD-regulated processes. Tumor hypoxia is induced by low oxygen levels triggering not only the formation of new blood vessels but also metastasis, as reviewed by Rankin et al. [84]. Hypoxia-induced metastasis is a complex process driven by several mediators. Widmer et al. demonstrated that the modulation of the hypoxia-inducible factor 1α (HIF1α) was sufficient to induce the dedifferentiation and phenotype switching of proliferative into invasive melanoma cells, referring to the AXL/MITF axis [85]. The stabilization of HIF1α under hypoxic conditions favored the invasive AXL^high^ state and the absence or low levels of HIF1α promoted the proliferative MITF^high^ phenotype. 

Changes in the growth factor environment yet present another mechanism, triggering the rapid interconversion of stem-like states, marked by CD271, CD133 [86] or CD271/CD133 [40]. We observed that cells with a double positive (CD271^+^/CD133^+^) as well as a double negative (CD271^−^/CD133^−^) phenotype were capable of re-establishing the cellular heterogeneity of the initial cell culture within three days. Moreover, a phenotype switch of CD271^+^/CD133^−^ to CD271^+^/CD133^+^ cells was induced in the absence of serum and FGF2, which was prevented in cells with a stable knockdown of CD271. Whether CD133^+^ stem-like cells indeed directly evolved from CD271^+^ cells or presented a differentiated progeny of yet another stem-like state is unknown. The shRNA-mediated down regulation of CD133 or CD271 in melanoma cells revealed that both markers are inversely correlated, suggesting that CD133 suppresses genes associated with CD271, e.g., AXL and FOXD3 and vice versa [40,86].

Although methods enabling the tracing of melanoma cell phenotypes in vitro exist, the monitoring of cues inducing phenotype switching within tumors, particularly in response to certain in vivo conditions, e.g., therapeutic interventions or hypoxia, remains challenging. Reporter systems may enable the in vivo tracing of phenotype switching of stem-like cells and their participation in tumor formation and metastasis. The combination of gene-specific 3′-UTR sequences fused to GFP provide insight into mechanisms maintaining stem-like cells (Figure 3a), e.g., regulated by microRNA binding. The FACS-enrichment (Figure 3b) of melanoma cells with a stable genomic integration of a 3´-UTR-CD271-GFP reporter yields in a pure GFP^+^ subset (Figure 3c, upper panels). However, only a minority of GFP^+^ cells is maintained upon culturing in two dimensional (2D) (Figure 3c, lower panels) or three dimensional (3D) systems (Figure 3d). Hence, the content of putative CD271^+^ stem-like cells within cell culture systems is tightly controlled by mechanisms balancing differentiation and dedifferentiation. 

In summary, these data suggest that CD271 likely controls melanoma cell migration and metastasis via phenotype switching which includes the suppression of MITF-targets. The comparison of MITF expression levels reveal that metastases and primary tumors featuring proliferative and invasive cell states (Appendix A) with no organ-specific prevalence (Appendix A). Hence, the expression of MITF/targets is suppressed during metastasis and rises during the formation of secondary tumors. In contrast, the expression of CD271 increases to facilitate metastasis, invasiveness and decreases once the disseminated cells arrived at distant organ sites. Moreover, a subset of metastases may permanently exist in a CD271^high^/MITF^low^ state featuring a high aggressiveness and metastatic capacity. The persistence of the CD271^high^/MITF^low^ state in vivo is unknown; however, therapeutic interventions potentially influencing the equilibrium of invasiveness and proliferation might select for the most favorable state (Appendix A) and consequentially foster specific metastatic routes, e.g., metastasis to the brain [87]. 

## 6. Signaling Mechanisms of CD271 in the Central Nervous System

CD271 is widely expressed in different cell types of the central nervous system (CNS), e.g., neurons and schwann cells, regulating a plethora of processes like survival and apoptosis, myelination and axonal regeneration in response to specific ligands like the neurotrophins (NGF, BDNF, NTF3, and NTF4). In addition, CD271 is expressed in several cellular compartments, is localized in the nucleus following neurotrophin-induced cleavage, the nuclear pore and shows membrane localization [76,88,89,90]. 

In the subventricular zone, the place of adult neurogenesis, CD271 labels a subset of progenitor cells, which are maintained in response to BDNF [91,92]. Consistently, the expression of CD271 is down regulated upon neuronal differentiation of progenitor cells in a rodent model system [92]. In addition, astrocytes gain expression of CD271 in response to injury and neuroinflammation that in turn is triggered in response to injury, seizure, toxic metabolites and tumor cells, as reviewed by Meeker et al. [93]. The activation of astrocytes, known as astrogliosis strongly determines the maintenance of tumor cells within the brain as reactive astrocytes secrete chemokines, cytokines and growth factors supporting tumor cell growth and migration, comprehensively reviewed by Sofroniew et al. [94]. The exact role of CD271 in astrogliosis is not well understood, although it might affect astrocyte proliferation [75]. CD271 lacks in a kinase-domain and regulates signaling processes in conjunction with co-receptors and/or the recruitment of signaling mediators that bind to the death domain of CD271 homodimers (Figure 4), as reviewed by Meeker et al. [93]. 

Signaling via CD271 (p75^NTR^) homodimers mediates the activation of NFĸB signaling, driving the expression of pro-survival genes via the recruitment of receptor-interacting serine/threonine-protein kinase 2 (RIP2). In the absence of the latter that competes with TRAF6 for binding to CD271, neurotrophins trigger the activation of c-jun-n-terminal kinase (JNK) to drive apoptosis in neurons [95] (Figure 4, first panel). In addition, the association of CD271 with tropomyosin-related kinases (TRKs) results in the generation of high-affinity neurotrophin binding sites and mediates cell survival in a mitogen-activated kinase (MAPK)-dependent manner via adaptor proteins SHC, GRB2, and SOS (Figure 4, second panel). Besides MAPK signaling, the CD271/TRK heterodimers signal trough PI3K and PLCγ [14]. Sortilin (SORT1), belongs to the vacuolar protein sorting 10 family (Vps10) and is mainly found within late endosomes derived from the trans-golgi network, controlling the anterograde trafficking of neurotrophin receptors and secretion of pro-BDNF [96]. The binding of neurotrophin precursors like pro-NGF or pro-BDNF to the CD271/SORT1 complex triggers apoptosis via adaptor proteins TRAF6, NRAGE and NRIF in neurons [97] (Figure 4, third panel), reviewed in references [16,98]. Moreover, CD271 forms a complex with Nogo receptor (RTN4R), MAG, and LINGO1 (not shown), inducing the release of RhoA from Rho GDP-dissociation inhibitor (RhoGDI) and subsequently activates the RhoA/ROCK signaling pathway. The latter in turn regulates the dynamic of the actin cytoskeleton (reviewed by Fujita et al. [99]), determining cell shape and motility via phosphorylation and activation of the Ser/Thr kinase LIM. Activated LIM in turn phosphorylates and inactivates the actin de-polymerization factor cofilin and mediates the inhibition of neurite outgrowth, reviewed in [93,99]. Furthermore, LIM activates the ERM proteins radixin (RDX), ezrin (EZR), and moesin (MSN) (Figure 4, fourth panel), which are important regulators of microvilli formation, cell adhesion, and membrane ruffling (reviewed by Tsukita et al. [100]) and potential drivers of cancer progression (reviewed by Clucas et al. [101]). 

However, CD271 signaling mechanisms show high context dependence, hence facilitate cellular properties in relation to the abundance of co-receptors, ligands, and recruited adaptor proteins that mediate signaling. Yet-identified signaling mechanisms in the CNS are more comprehensively discussed by Roux and Barker [102]. 

## 7. Signaling of CD271 in Melanoma and Other Cancer Types 

During central nervous system (CNS) development, the NC gives rise to several differentiated progeny, besides melanocytes like glia cells, which in turn give rise to astrocytes. Several lines of evidence suggest that particular cancer entities have adopted molecular features used by NC cells (reviewed by Maguire et al. [103]; Appendix A). 

The first evidence that CD271 consequentially determines melanoma cell behavior was provided by Shonukan et al. who demonstrated the specific interaction of CD271 with the actin cytoskeleton via the actin-bundling protein fascin (FSCN1). This interaction facilitates the migration of melanoma cells in a NGF and pro-NGF dependent manner [104]. Moreover, Marchetti et al. observed the activation of heparanase (HPSE1) expression in melanoma triggered by NGF and CD271 in a dose-dependent manner and even in the absence of TRKA [6,105,106]. HPSE1 promoted melanoma cell invasion and in turn was reported to induce PI3K/AKT signaling and probably promote tumor cell migration modulated by PLEKHA5, which specifically interacts with phosphoinositides [107]. Furthermore, Restivo et al. suggested that the binding of NGF to the complex of CD271 and TRKA triggered cell detachment and cleavage of CD271, yielding in the CD271-ICD that, in turn, induced the “growing” to “going” switch [53]. Moreover, the cleavage of CD271 by ADAM10 or ADAM17 controls the formation of the intracellular domain fragment (ICD) or carboxyterminal fragment (CTF), respectively. Both fragments mediate resistance to apoptosis mediated by TRAIL (CTF) or triggered by anoikis (ICD) in breast cancer and/or HNSCC [108,109] (Figure 5b, Appendix A). The malignant transformation of glia cells causes the development of astrocytoma including glioblastoma, the major forms of primary brain tumors in adulthood. Although astrocytes lack expression of CD271, several processes within the CNS like astrogliosis increase the level of CD271 on astrocytes. Like for melanoma, the contribution of CD271 to the malignant transformation of astrocytes is completely unknown. Subsets of glioblastoma and astrocytoma (Figure 5a, not shown) as well as primary tumors and BM of melanoma (Figure 5a, left panels, and Figure 1d) and lung squamous cell carcinoma (LSCC) (Figure 5a, right panels and Appendix A) feature a high expression of CD271 [110,111], as reviewed by Wrensch et al. [112]. Concordantly, high levels of CD271 drive migration and invasion of glioblastoma cells, mediated via the PDZ and LIM-domain protein PDLIM1 [111,113], (Figure 5b, Appendix A). The phosphorylation of CD271 at S303/S304 by protein kinase a (PKA) in turn promoted glioblastoma invasiveness and the translocation of CD271 to lipid rafts in neurons and potentially in glioblastoma [113,114]. 

In addition, the expression of CD271 was associated with increased migration/invasion and metastasis of head and neck cancer, hypopharyngeal cancer, and oral squamous cell carcinoma [115,116,117] and was associated with poor prognosis of LSCC [118] and urothelial carcinoma (TCGA, human protein atlas, Appendix A). Strikingly, the tumor formation of engrafted hypopharyngeal cancer and melanoma cells was blocked by a humanized anti-CD271 antibody [119]. In contrast, a high expression of CD271 in medulloblastoma, esophageal squamous cell carcinoma, gastric cancer (Appendix A), and prostate cancer [120] was associated with a favorable tumor state. 

In summary, CD271 defines properties of tumor cells of several cancer entities, often associated with a poor outcome. Additional studies are needed to uncover molecular mechanisms that are common and distinct to specific cancer entities. 

## 8. Therapeutic Targeting of CD271

Signaling mechanisms of CD271 in melanoma and other tumor entities are still not well understood, making the specific targeting challenging. As CD271 regulates survival and apoptosis of neurons in conjunction with co-receptors and in dependence of the availability of ligands within the CNS, the direct targeting of either CD271 or neurotrophins seems impossible. Nevertheless, commercially available small molecule drugs like non-peptide inhibitors preventing NGF from binding to CD271; PD90780, Ro 08-2750, and peptide inhibitor NTR368 have not yet been tested in vivo. However, a non-peptide ligand of CD271, LM11A-31 was administered to patients suffering from mild to moderate Alzheimer’s disease in a clinical trial (NCT03069014). The inhibitor potentially inhibits CD271-mediated apoptosis of neurons.

Yet, the most intriguing finding by Ngo et al. suggests that the blocking of CD271 by neutralizing antibodies prevents melanoma metastasis in vivo [79]. Recently, Morita et al. reported that a humanized anti-CD271 antibody was sufficiently blocking the tumor growth of MeWo graft models [119]. However, the systemic application of CD271 neutralizing antibodies like inhibitors might negatively interfere with processes within the CNS. The targeting of CD271 is a promising route potentially preventing melanoma from metastasis. Two clinical pilot studies investigating whether T cells genetically engineered to express NGFR/CD271 improve the recognition and eradication of melanoma cells in patients with stage III–IV melanoma, started in 2012 and 2013 (NCT01955460, NCT01740557). It is likely that the adoptive transfer of autologous tumor infiltrating lymphocytes (TIL) is superior to other therapeutic interventions due to higher selectivity [121].

## 9. Conclusions and Remarks

The nerve growth factor receptor CD271 regulates the survival and apoptosis of neurons within the CNS and basic properties of melanoma cells like tumorigenicity, plasticity, and self-renewal. Moreover, the molecular programs determining the migratory capacity of NCSCs likely drive the migration and metastasis of melanoma cells. The expression of CD271 increases in response to therapeutic drugs, inflammation, and stress and mediates drug resistance. Hence, CD271 likely acts as a key mediator of therapy-induced invasion and metastasis in melanoma and other cancer types. Although the molecular programs controlling the survival, apoptosis, and migration of cells in the CNS are well understood, the signaling mechanisms of CD271 in cancer remain elusive due to the high diversity of modes of signaling. The latter is strongly determined by the levels of co-receptors, ligands adaptor proteins that are recruited to the cytoplasmic domain of CD271. CD271-expressing brain-metastatic cells exhibit a more aggressive phenotype and likely benefit from close proximity to reactive astrocytes. All these findings strongly motivate the therapeutic targeting; however, the understanding of the signaling mechanisms that are directly controlled by CD271 is a prerequisite for the development of therapeutic drugs.

## Figures and Tables

**Figure 1 cancers-12-02460-f001:**
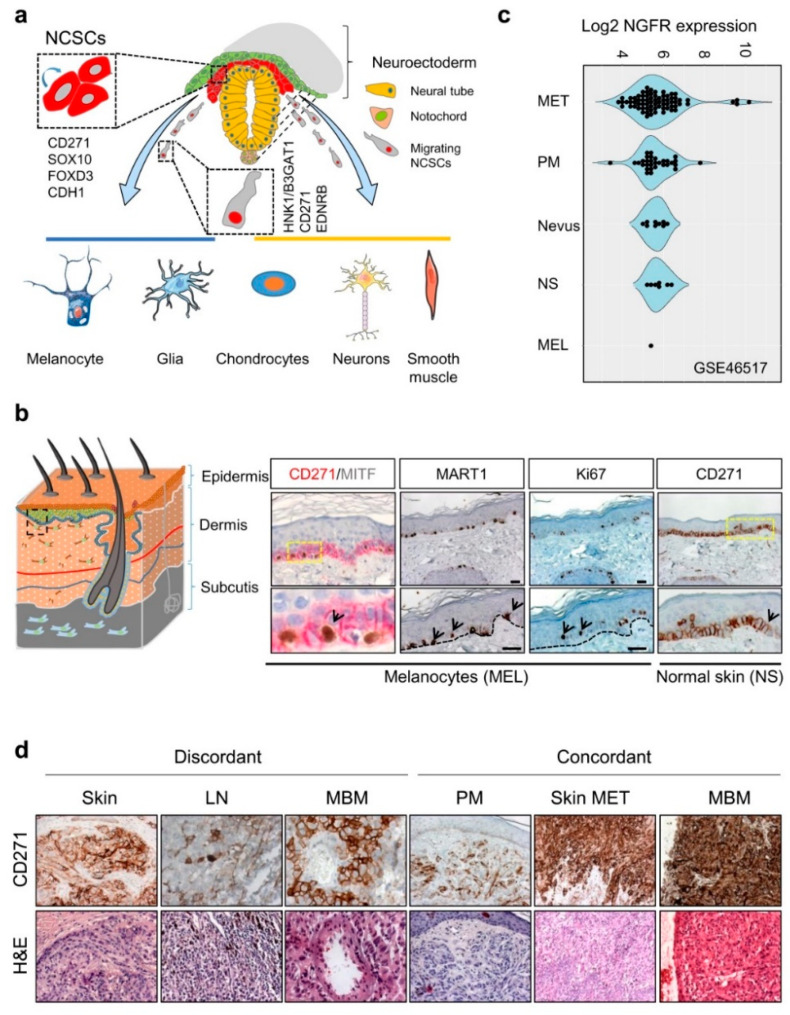
CD271 in development and melanoma. (**a**) Simplified schematic representation of neural crest cell delamination and migration and markers of migrating (HNK1/B3GAT1) and non-migrating (CD271, SOX10, FOXD3, CDH1) neural crest stem cells (NCSCs). The regulated differentiation of NCSCs give rise to melanocytes, glia cells, chondrocytes, neurons and smooth muscle cells. (**b**) Schematic representation of the skin layers (left) and location of CD271 expressing transient-amplifying keratinocytes (CD271^+^/MITF^−^/MART1^−^) and melanocytes (CD271^+^/MITF^+^/MART1^+^, black arrows), right panels. Proliferative cells/melanocytes, aremarked by Ki67 (black arrows). (**c**) The comparative analysis of different melanoma progression stages (Nevus, primary melanoma; PM, and metastases; MET), normal skin (NS), and melanocytes (MEL) of study GSE46517 revealed increased levels of NGFR/CD271 by trend. (**d**) The analysis of discordant and concordant metastases revealed the expression of CD271 in metastases of different organs (skin, lymph node; LN and brain; MBM) and the maintenance of CD271 among concordant samples.

**Figure 2 cancers-12-02460-f002:**
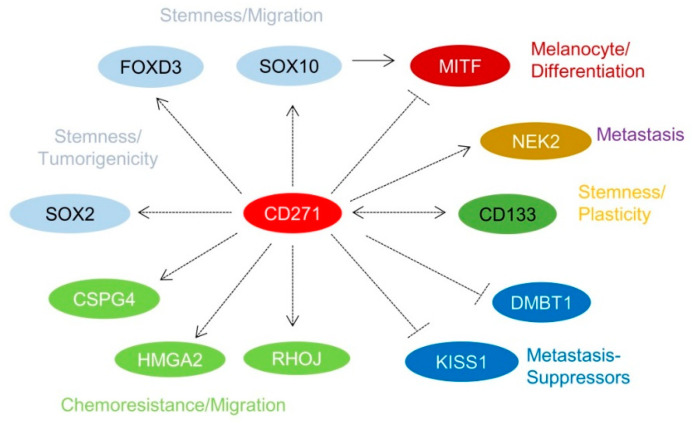
CD271 serves as regulator of basic properties of melanoma cells. Our knockdown study (GSE52456) uncovered several CD271-associated genes; some of them are depicted. Among them are mediators of stemness and migration of NCSCs, tumorigenicity, and chemoresistance, migration, and plasticity.

**Figure 3 cancers-12-02460-f003:**
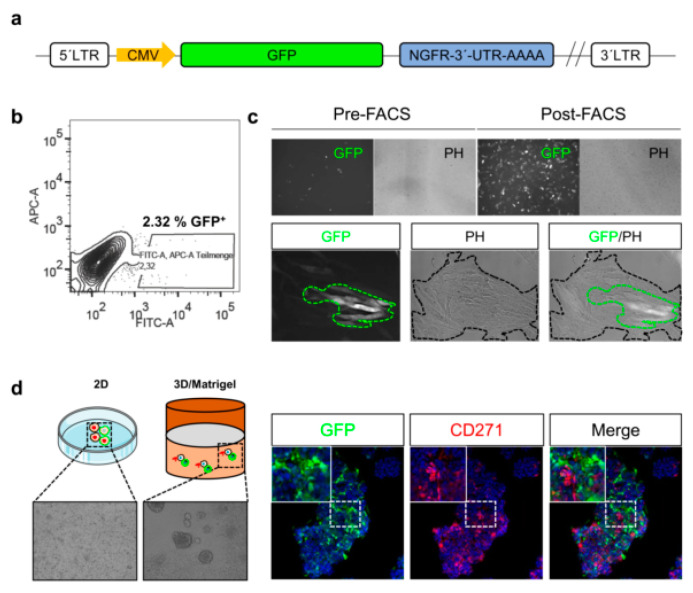
CD271 regulates phenotype switching in melanoma. (**a**,**b**) A reporter plasmid containing the NGFR/CD271-specific 3′-UTR sequence fused to GFP (green fluorescent protein) enanled theindirect tracing and isolation of NGFR/CD271 expressing melanoma cells. a reporter plasmid (**c**) Cells with a stable integration of the reporter were enriched via FACS (fluorescence activated cell sorting) (upper panels). The absence of GFP indicates the suppression of NGFR transcription or mRNA degradation e.g., via miRNAs. GFP expression was lost over time but maintained in a cellular subset (**c**, lower panels). (**d**) GFP^+^ cells participated in the formation of three-dimensional (3D) spheres in matrigel.

**Figure 4 cancers-12-02460-f004:**
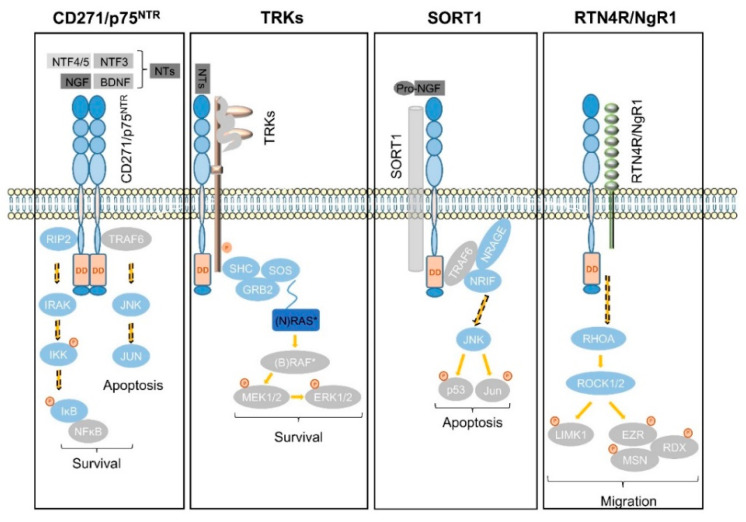
Signaling mechanisms of CD271. First panel: in the absence or low expression of co-receptors, CD271 form homotetramers (homodimers are shown) and controls either cell survival via a RIP2-mediated activation of NFĸB and downstream pro-survival genes or apoptosis via TRAF6. The latter competes with RIP2 for binding to CD271 and activates JNK (c-Jun N-terminal kinases)/JUN signaling. Second panel: The activation of the typical mitogen-activated protein kinase pathway (MAPK) is triggered by binding of neurotrophins to the complex of CD271 and tropomyosin-related kinases (TRKs), leading to signaling processes mediated by ERK1/2 (extracellular-regulated kinases) or PI3K (phosphatidylinositol 3-kinases) promoting proliferation and cell survival. Third panel: the aggregation of CD271 with sortilin triggers apoptosis in response to neurotrophin precursors, e.g., pro-NGF. Apoptosis is induced via the recruitment of TRAF6 (tnf-receptor associated factor 6), NRAGE/MAGED1 and NRIF to the dead domain (DD) of CD271 via c-jun n-terminal-kinase (JNK) or via activation of NFĸB. Fourth panel: CD271 triggers the activation of RHOA (RAS-homolog family member A) and RHO-kinases which in turn activate LIM-kinase (LIMK1) and ERM proteins (EZR; ezrin, RDX, radixin and MSN; moesin) to control cell migration in conjunction with nogo receptor/RTN4R (reticulon 4 receptor).

**Figure 5 cancers-12-02460-f005:**
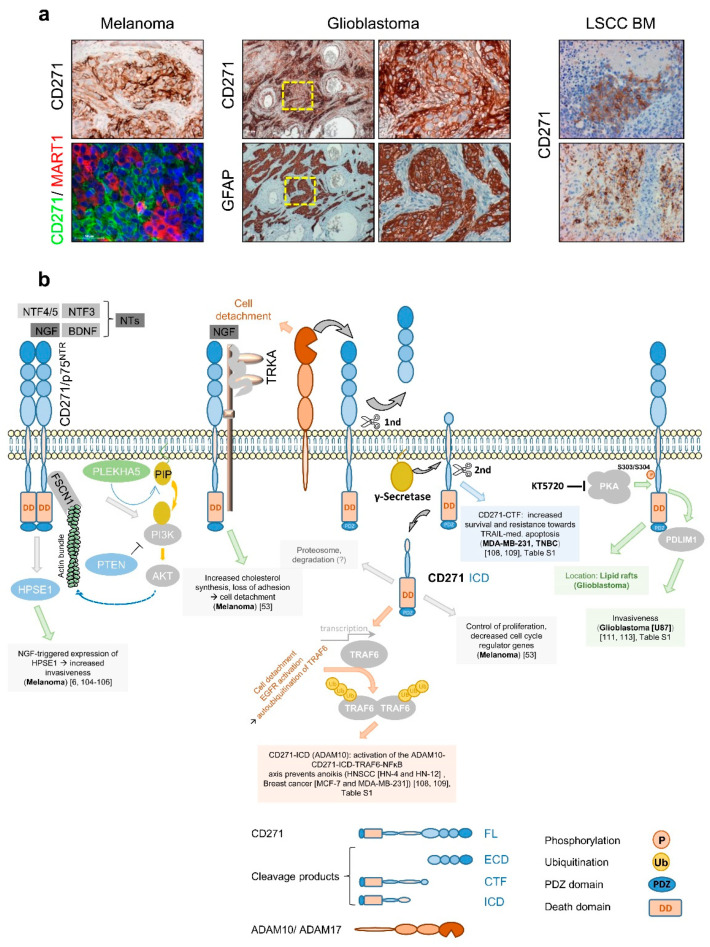
Expression and consequence of CD271 in melanoma and beyond. (**a**) CD271 labels a subset of tumor cells that feature a stem-like phenotype, lacking expression of melanoma/melanocyte specific markers like MART1 (first panels). The CD271 expression in GFAP^+^ glioblastoma cells drives invasiveness and is associated with poor prognosis (center panels). Like in glioblastoma, the expression of CD271 defines a proliferative state and is associated with poor prognosis of lung squamous cell carcinoma (LSCC) and is found in BM of LSCC (right panels). (**b**) Intracellular signaling processes of CD271 are triggered by neurotrophin binding to CD271 homodimers, triggering the binding to actin bundles via fascin (FSCN1), the activation of heparanase (HPSE1), or induce PI3K/AKT-signaling and cell migration modulated by PLEKHA5. The binding of NGF to the complex of CD271/TRKA was suggested to increase cholesterol synthesis and cell detachment via formation of the CD271-ICD fragment. The latter in turn decreased the levels of cell cycle regulators and triggered the phenotype switch from proliferation (“growing”) to invasion/migration (“going”). The intracellular processing of CD271 by ADAM-mediated cleavage steps leads to the formation of a CTF that is additionally processed by γ-secretase to form the ICD fragment. The CTF was demonstrated to mediate increased survival and resistance to TRAIL-mediated apoptosis in MDA-MB-231 breast cancer cells. The ICD in turn triggered the activation of the ADAM10-CD271-ICD-TRAF6-NFĸB axis to prevent anoikis in head and neck squamous cell carcinoma (HNSCC) and breast cancer cells by the activation of epidermal growth factor receptor (EGFR) signaling and degradation of TRAF6. In glioblastoma, the phosphorylation of CD271 at S303 within the juxtamembrane domain triggered the binding of PDLIM1 and increased invasiveness or the potential translocation into lipid rafts blocked by the PKA inhibitor KT5720.

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
