# Peer review of "Decoding the Role of CD271 in Melanoma"

_cancers, 2020, doi:10.3390/cancers12092460_

Round 1
Reviewer 1 Report
In the present review entitled: “Decoding the role of CD271 in melanoma” the Authors have discussed the most recent knowledge and findings about CD271-associated mechanisms in melanoma cells and illustrate the role of CD271 for melanoma cell migration and metastasis, phenotype-switching, resistance to therapeutic interventions and the maintenance of a NCSC-like state.
Yet, the review encompasses a bright spectrum of the literature which makes the manuscript very conclusive to some extent complex.
However, some minor suggestions/additions might be relevant and interesting for the present manuscript.
The review is worth publication, but before, I would recommend to short the entire text or to re-define which parts of the Ms are really important to include in answering the major role of the CD271 (see the goal above).
In the chapter 2.1. referring to the figure 1 and the corresponding text passages, it seems that there is no complete compliance. For instance the abbreviations for e.g. HNK1(=B3GAT1) and EDNRB did not have been included in the text as well as below the figure appropriately. Even if the Authors have mentioned the abbreviations later it would be more clear, to rearrange the sentences correspondingly before. In some cases were some abbreviations clarified repetitive.
E.g. in line 83 are fonts bold or regular for naming the figures ?
Overall, please check for errors/typing, scale bars were failing. The description of the figure 2 is not consistent.
In general, there are many original data from the Authors which it would be worth to see in the original work instead… It makes the Review to comprehensive and distracts the reader, too.
Figure 6 should be adopted for publication and shown in a better quality.
Author Response
Dear Reviewer 1,
Thank you very much for the valuable comments and advice. We changed the manuscript as follows:
- We changed/nourished the indication of HNK1/B3GAT1 in the text and in figure 1
- We reorganized the whole manuscript. To this end, we removed the chapter “Melanocytic Lineage Commitment”, which is not directly within the focus, removed original data from figures 2 and 5 and removed figure 4 completely.
- We changed the formatting of the figure´s name, now fonts of figures are depicted normal
- We streamlined the manuscript were applicable and removed typos
- Figure 6 is now embedded with a better resolution (600 dpi)
The structure of the manuscript in the present state is as follows:
Page 1. Line 27: Introduction, 1
Page 2. Line 63: CD271 in development, 2 ; line 64: Neural crest stem cells
Pages 3. and 4 contain figure 1 and the updated figure legend
The section “CD271 in the skin and development of melanoma” is now section 3, starting in line 108 on page 4. Now figure 2 only contains the CD271-associated network on page 5.
The section “Identification of CD271-associated genes” is now section 4, starting in line 187 on page 6, followed by section 5 “CD271 in migration, metastasis and cellular plasticity” in line 211.
The initial figure 3 has now been replaced by figure 5d-f which is now figure 3a-c on page 8.
The section “Signaling mechanisms of CD271 in the central nervous system” is now section 6, starting in line 334 on page 9 and is now associated with figure 4 (formerly figure 3) on page 10.
The section “Signaling of CD271 in melanoma and other cancer types” is now section 7, starting in line 397 on page 11, associated with figure 5 (formerly figure 6), followed by 8 and 9, “Therapeutic targeting of CD271” and “Conclusion and Remarks“, starting in lines 459 and 480 on page 13, respectively.
We hope that these changes improved the manuscript. For a better survey, the manuscript is submitted with accepted and non-accepted track changes.

Reviewer 2 Report
I found it a little difficult to decide what to make of this work. Several assertions are written without reference to the analysis or without reference to the bibliography. The writing often lacks clarity and sharpness, and several sections are poorly organized or do not flow with the rest of the paper (e.g., Line 52 CSCs is written for the first time and the abbreviation details were not written in this paper, line 206 Figure 1c is wrong, line 298 Figure 3a isn’t presence, and so on).
Author Response
Dear reviewer 2, the manuscript has been reorganized and streamlined and the erroneous information has been revised.

Reviewer 3 Report
The authors provided a review about the role of CD271, the topic is interesting the paper is well organized, but some revisions are required since it is presented in vary shabby way, Especially the iconography
- Please correct the keywords, I think that there are some refuses.
- In the text some gene name acronyms are not defined, please define
- At line 144, I think that the sentence refers to Figure 1C, please check and correct
- At line 148, are you sure that the sentence refers to figure S1b-c? Because that figure shows the expression of different proteins in the skin, please check
- At line 206, maybe the sentence refers to figure 1d, please check
- At line 236, I think that the figure showing the expression of SOX10 is the S1c, instead of S1b, please check
- At line 404, MITF expression is shown in figure 4d, instead of 4c, please check.
- In figure 6a is reported the expression of CD271 in lung cancer, but it is not mentioned in the text, please remove the panel showing the expression of CD271 in lung cancer.
- I suggest reformulating the sentence from line 298 to 351, because in the figure 4 (referring to that sentence) is shown also genes involved in DNA repairing, but they are not mentioned into the text.
Author Response
Dear Reviewer 3,
Thank you very much for the valuable comments and advice. We changed the manuscript as follows:
Please correct the keywords, I think that there are some refuses.
Has been revised.
In the text some gene name acronyms are not defined, please define
We agree and provided full gene names for B3GAT1 and EDNRB.
At line 144, I think that the sentence refers to Figure 1C, please check and correct.
The labeling in the figure legend was misleading, this has been now corrected.
At line 148, are you sure that the sentence refers to figure S1b-c? Because that figure shows the expression of different proteins in the skin, please check
Yes indeed, the sentence refers to the immunohistochemistry of BCL2, c-KIT, CK14, MITF and CD271 in the skin, showing the presence of BCL-2/c-KIT/MITF positive melanocytes. However, the sentence does not refer to figure S1c, which is now corrected.
At line 206, maybe the sentence refers to figure 1d, please check
Yes, we agree. The sentence (now line 171) refers to figure 1c, showing the violin plot depicting the levels of NGFR/CD271 in different melanoma progression stages and to figure 1d showing the staining of CD271 in melanoma metastases.
At line 236, I think that the figure showing the expression of SOX10 is the S1c, instead of S1b, please check.
We agree and changed this accordingly, now in line 191.
At line 404, MITF expression is shown in figure 4d, instead of 4c, please check.
We agree, however due to a reorganization of the manuscript, we completely removed figure 4 and added the citation of our manuscript showing the data in line 278.
In figure 6a is reported the expression of CD271 in lung cancer, but it is not mentioned in the text, please remove the panel showing the expression of CD271 in lung cancer.
The figure 6 is now figure 5. In figure 5a, right panel, we depict the presence of CD271 in lung cancer brain metastases which has now been added to the main text (line 421-422) and changed in the figure legend.
I suggest reformulating the sentence from line 298 to 351, because in the figure 4 (referring to that sentence) is shown also genes involved in DNA repairing, but they are not mentioned into the text.
The figure 4 has been withdrawn, our publication showing these data added to the main text.
In addition, we changed the manuscript as follows:
The structure of the manuscript in the present state is as follows:
Page 1. Line 27: Introduction, 1
Page 2. Line 63: CD271 in development, 2 ; line 64: Neural crest stem cells
Pages 3. and 4 contain figure 1 and the updated figure legend
The section “CD271 in the skin and development of melanoma” is now section 3, starting in line 108 on page 4. Now figure 2 only contains the CD271-associated network on page 5.
The section “Identification of CD271-associated genes” is now section 4, starting in line 187 on page 6, followed by section 5 “CD271 in migration, metastasis and cellular plasticity” in line 211.
The initial figure 3 has now been replaced by figure 5d-f which is now figure 3a-c on page 8.
The section “Signaling mechanisms of CD271 in the central nervous system” is now section 6, starting in line 334 on page 9 and is now associated with figure 4 (formerly figure 3) on page 10.
The section “Signaling of CD271 in melanoma and other cancer types” is now section 7, starting in line 397 on page 11, associated with figure 5 (formerly figure 6), followed by 8 and 9, “Therapeutic targeting of CD271” and “Conclusion and Remarks“, starting in lines 459 and 480 on page 13, respectively.
We hope that these changes improved the manuscript. For a better survey, the manuscript is submitted with accepted and non-accepted track changes.
